



# Employing the Generalized Pareto Distribution to Analyze Extreme Rainfall Events on Consecutive Rainy Days in Thailand's Chi Watershed: Implications for Flood Management

Tossapol Phoophiwfa[1], Prapawan Chomphuwiset[1], Thanawan Prahadchai[2], Jeong-Soo Park[2], Arthit Apichottanakul[3], Watchara Theppang[4], and Piyapatr Busababodhin[1,*]

[1]Digital Innovation Research Cluster for Integrated Disaster Management in the Watershed, Mahasarakham University, Kantharawichai, Maha Sarakham 44150, Thailand.
[2]Department of Statistics, Chonnam National University, Gwangju 61186, Korea.
[3]Department of Production Technology, KhonKean University, Khon Kean, 44000 Thailand.
[4]Buengkan Provincial Agricultural Extension Office, Buengkan 38000, Thailand.

**Correspondence:** *Piyapatr Busababodhin (piyapatr.b@msu.ac.th)

**Abstract.** Extreme rainfall events in the Chi watershed of Thailand have significant implications for the safe and economic design of engineered structures and effective reservoir management. This study investigates the characteristics of extreme rainfall events in the Chi watershed, Northeast Thailand, and their implications for flood risk management. We apply extreme value theory to historical maximum cumulative rainfall data for consecutive rainy days from 1984 to 2018. The Generalized Pareto Distribution (GPD) was used to model the extreme rainfall data, with the parameters estimated using Maximum Likelihood Estimator (MLE) and Linear Moment Estimator (L-ME) methods based on specific conditions. The goodness-of-fit tests confirm the suitability of the GPD for the data, with p-values exceeding 0.05. Our findings reveal that certain regions, notably Udon Thani, Chaiyaphum, Maha Sarakham, Tha Phra Agromet, Roi Et, and Sisaket provinces, show the highest return levels for consecutive 2-day (CONS-2) and 3-day (CONS-3) rainfall. These results underscore the heightened risk of flash flooding in these regions, even with short periods of continuous rainfall. Based on our findings, we developed 2D return level maps using the Q-geographic information system (Q-GIS) program, providing a visual tool to assist with flood risk management. The study offers valuable insights for designing effective flood management strategies and highlights the need for considering extreme rainfall events in water management and planning. Future research could extend our findings through spatial correlation analysis and the use of copula functions. Overall, this study emphasizes the importance of preparing for extreme rainfall events, particularly in the era of climate change, to mitigate potential flood-related damage.

## 1 Introduction

The distribution of rainfall and atmospheric fluctuations are directly impacted by changes in climate, which have significant implications for water resource management and hydrology. The Northeast region of Thailand is particularly susceptible to frequent flooding, which is often caused by a combination of local conditions, natural variations, and human actions. Unfortunately, this issue shows no signs of abating, and it continues to escalate in severity. The Northeast region of Thailand is home





to over 63 million rai (1 rai = 1,600 square meters) of agricultural land, a significant proportion of which still experiences water shortages, droughts, and flooding. Over the past three decades, water shortages have affected 57 provinces (or 75% of the country), 525 districts (or 60% of the total districts), 3,321 sub-districts (or 46% of the total), and 24,900 villages (or 33% of the total villages) in Thailand, causing extensive damage. On average, 9.71 million people are affected each year, representing

about 15% of the total population who suffer from drought annually. Additionally, an average of 2.571 million rai of farmland is damaged each year, leading to an average loss of 661 head of livestock. The total cost of damage amounts to 656.62 million baht (or 17.57 million US dollars) per year. Furthermore, the Northeast region has experienced seven major floods over the years 1983, 1995, 1996, 2002, 2006, and 2011, which have caused significant damage to both human life and property, making it difficult to assess the total cost of damage incurred (Gale and Saunders, 2013; Singkran, 2017; Meteorological, 2021). Gale

and Saunders (2013) identified the causes of the major floods that occurred in Thailand in 2011 and presented forecasts for future flooding. Their research indicates that unless flood defenses and management practices are improved, there is a high likelihood of more flooding occurring within the next two to three decades.

According to the Thai Meteorological Department's report in 2006 (Meteorological, 2021), flood conditions in the Chi Watershed occur 2-3 times a year. Various studies have also indicated that the area is prone to frequent flooding (Kunitiyawichai

et al., 2011; Arunyanart et al., 2017). Flooding in the Chi Watershed takes on many different forms, including but not limited to overflowing riverbanks in provinces such as Chaiyaphum, Khon Kaen, and Roi Et; wild water flows in Chaiyaphum, Khon Kaen, and Roi Et; and mudslides in Kalasin and Chaiyaphum Provinces. The watershed has also experienced severe flooding in various areas, such as Roi Et, Kalasin, and Khon Kaen Provinces. The Chi Watershed is susceptible to flooding due to several factors. First, heavy rainfall resulting from the influence of the southwest and northwest monsoons and depressions

from the South China Sea often occurs in the watershed area. Second, the upstream area of the watershed, where the Chi River originates, is characterized by mountainous terrain with high slopes and has experienced significant deforestation. Third, the lower part of the watershed, particularly in Roi Et and Ubon Ratchathani Provinces, is a plain where multiple rivers converge and is the point where the Chi River meets the Mun River before flowing into the Mekong River. This creates drainage issues for the watershed area. Fourth, water management in large reservoirs poses a challenge during the rainy season, as some years

require significant amounts of water to be drained due to the high levels of annual rainfall and water discharge from nearby reservoirs (Meteorological, 2021). Given these challenges, effective water management during the flooding and drought seasons is critical. Numerous studies have applied mathematical and statistical theories to address these issues, such as those conducted by (Bhakar et al., 2006; Noymanee and Theeramunkong, 2019; Suksawang, 2012; Hung et al., 2009; Dutta et al., 2003)

It is well-known that floods occur on average every several years, as supported by numerous studies. In this context, Coles

(2001) introduced the concept of extreme value theory, which focuses on studying the maximum and minimum occurrences in a dataset. These extreme values are typically located at the tail of the distribution and are often disregarded in analysis or modeling due to their perceived complexity and low number. However, extreme value theory provides a framework to better understand and model such events. Extreme analysis is a method employed to assess the severity of natural phenomena, encompassing factors such as maximum-minimum rainfall, temperature extremes, maximum-minimum wind speeds, and more.

In their study, (Busababodhin and Kaewmun, 2015; Pangaluru et al., 2018; Wang and Xuan, 2020) developed an extreme value





model to analyze the probability of extreme events using data from Thailand. They also explored methods for selecting the most suitable extreme value model and determining return periods and return levels. Bhakar et al. (2006) studied the analysis of the frequency of one day maximum rainfall and two to five days consecutive maximum rainfall at Banswara District in southern Rajasthan of India. Three distributions, the normal, log-normal and Gumbel distributions, were used in the analysis

for this data and compared with the Chi-square value, the results showed that the Gumbel distribution was the best fit for the region and it was taken for the return level associated with return periods varying from 2 to 100 years. Several studies have investigated the frequency of maximum consecutive days rainfall, and they support the use of extreme value distribution. These studies employed maximum likelihood estimation (MLE) and verified model suitability using tests such as the Kolmogorov-Smirnov (KS) test and Anderson-Darling (AD) test. Examples of these studies include (Kwaku and Duke, 2007; Patel et al.,

2011; Manikandan and Kumar, 2015; Sabarish et al., 2017)

The current study aimed to fill the research gap by examining the consecutive days' maximum rainfall data in Thailand. This data set was chosen due to the frequent occurrence of flooding caused by continuous heavy rainfall. To the best of our knowledge, no previous studies have been conducted on this specific type of data in Thailand. In this study, we aim to identify critical areas along the Chi Watershed and evaluate their severity for use in planning, resolving flooding, and pre-evaluating

damage. To achieve this, we applied the non-stationary Generalized Pareto Distribution (NS GPD) models on the maximum cumulative rainfall data observed for consecutive rainy days of 2, 3, 4, 5, 6, and 7 days at 18 stations along the Chi Watershed in the northeastern region of Thailand. Section 2 provides an overview of the data and climatology of the Chi watershed in Thailand. Section 3 describes the materials and methods used in the study, including the NS GPD modelling, which considers five models. In Section 4, the results of the study are presented, including isopluvial maps of the return levels and their changes

over time, which were predicted from the best model. Discussions are provided in Section 5, followed by a conclusion in Section 6. Technical specifics, tables, and figures are included in the Supplementary Materials.

## 2 Data

In this study, we analyzed the maximum cumulative rainfall on consecutive rainy days (2, 3, 4, 5, 6, and 7) data observed by the Thai Meteorological Department (TMD) (Meteorological, 2021) from 1984 to 2022. The rainfall data ranged from 115.0

to 330.0 mm, with an average range of 17.7 to 114.44 mm for all stations. Descriptive statistics for the maximum cumulative rainfall on consecutive rainy days (2, 3, 4, 5, 6, and 7) for some stations are presented in Table 1, where $N^*$ is the number of consecutive rainfalls between 1984 and 2022. We used the maximum cumulative rainfall for consecutive rainy days to select the number of consecutive rainy days for analysis as in Eq. (1).

$$\text{CONS-n} = \sum_{i=1}^{n}(X_i), \tag{1}$$

where $X_i > 0$; $X_i$ is rainfall on consecutive rainy days, such as in the case $n = 2$ then CONS-2 $= X_1 + X_2$ when $X_1, X_2$ is rainfall on one and two consecutive rainy days, respectively. Figure 3 displays the density curves for cumulative rainfall on





consecutive rainy days, indicating that all stations are positively skewed and have heavy tails. Further details are provided in the supplementary material (SM).

Figure 1 displays the locations of the 18 meteorological stations situated along the Chi watershed in the northeastern region

of Thailand, covering 12 provinces. The detailed latitude and longitude of these stations are provided in Table S1 (moved to the Supplementary Materials). The Chi watershed falls within the tropical area, spanning between latitudes $13^o00'$ to $18^o00'$ and longitudes $101^o00'$ to $105^o00'$.

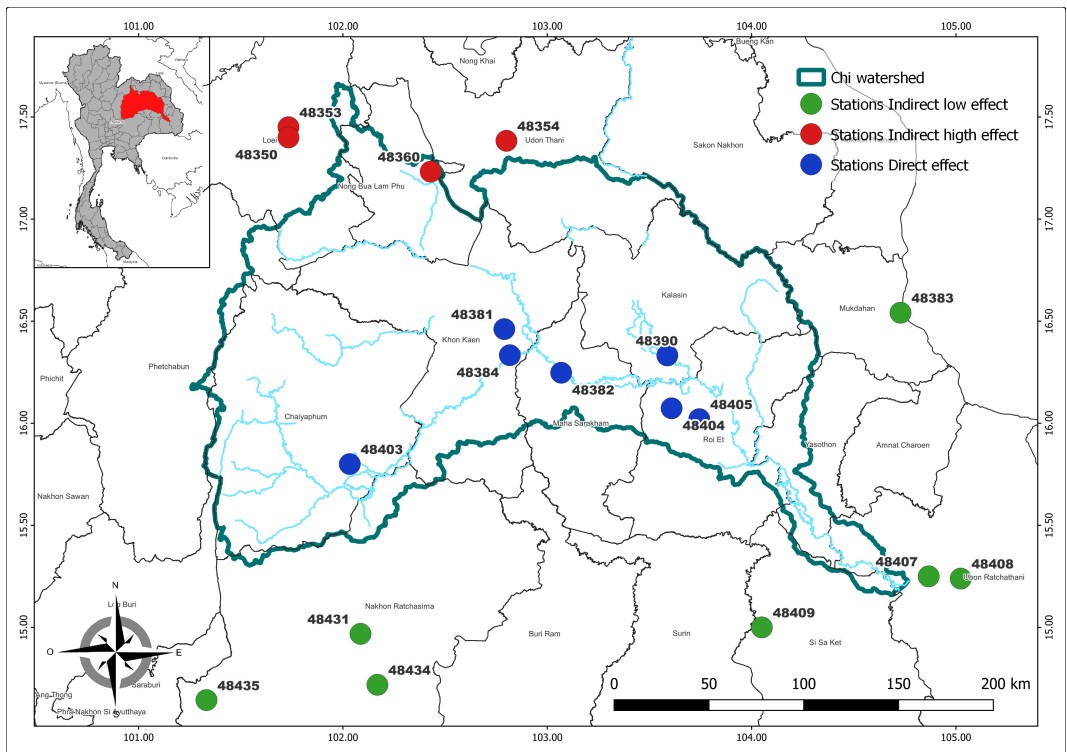

**Figure 1.** Location of all 18 meteorological stations along the Chi Watershed in northeastern region of Thailand.





**Table 1.** Descriptive Statistics of Maximum Cumulative Rainfall on Consecutive Rainy Days (2, 3, 4, 5, 6 and 7 days) for Selected Stations in Northeast Thailand, with Top Three Maximum Values Highlighted. $N^*$ Represents the Number of Consecutive Rainy Days (mm.).

| Data | Station | N* | Min | Mean | Median | Max | Data | Station | N* | Min | Mean | Median | Max |
|------|---------|-----|------|-------|--------|--------|--------|---------|----|-------|--------|--------|--------|
| CONS-2 | 48353 | 377 | 0.20 | 18.05 | 10.80 | 115.00 | CONS-5 | **48353** | **91** | **5.50** | **54.03** | **45.50** | **232.40** |
| | 48381 | 390 | 0.11 | 20.12 | 13.16 | 155.80 | | 48381 | 87 | 5.02 | 62.13 | 53.80 | 216.60 |
| | **48384** | **403** | **0.02** | **21.79** | **14.10** | **194.00** | | 48384 | 89 | 6.60 | 69.32 | 61.01 | 188.80 |
| | 48382 | 436 | 0.02 | 23.29 | 15.55 | 164.80 | | 48382 | 68 | 3.60 | 79.95 | 68.95 | 228.00 |
| | 48390 | 180 | 0.02 | 21.93 | 14.25 | 171.40 | | **48390** | **34** | **9.20** | **76.55** | **62.90** | **228.40** |
| | **48403** | **367** | **0.02** | **20.72** | **12.20** | **201.60** | | 48403 | 70 | 3.60 | 52.11 | 37.00 | 211.00 |
| | 48405 | 399 | 0.02 | 23.26 | 15.70 | 135.10 | | 48405 | 59 | 2.20 | 60.75 | 62.30 | 182.10 |
| | 48404 | 401 | 0.21 | 21.52 | 14.30 | 141.80 | | **48404** | **73** | **7.90** | **68.50** | **60.10** | **228.30** |
| | 48407 | 364 | 0.11 | 22.69 | 13.80 | 129.40 | | 48407 | 74 | 8.00 | 70.02 | 70.00 | 227.60 |
| CONS-3 | 48353 | 218 | 0.21 | 33.70 | 23.90 | 147.80 | CONS-6 | **48353** | **63** | **11.20** | **68.25** | **51.30** | **304.20** |
| | **48381** | **199** | **0.80** | **36.74** | **28.30** | **273.60** | | 48381 | 50 | 16.60 | 76.58 | 68.85 | 159.30 |
| | 48384 | 220 | 0.80 | 34.66 | 27.65 | 174.70 | | **48384** | **47** | **10.30** | **77.82** | **62.90** | **289.80** |
| | 48382 | 210 | 0.60 | 38.95 | 29.45 | 189.10 | | 48382 | 46 | 12.00 | 84.72 | 70.65 | 210.40 |
| | 48390 | 90 | 0.32 | 30.76 | 22.55 | 151.30 | | **48390** | **30** | **11.50** | **100.20** | **73.10** | **303.00** |
| | 48403 | 231 | 1.00 | 36.65 | 28.30 | 163.40 | | 48403 | 61 | 9.00 | 72.67 | 73.90 | 152.20 |
| | **48405** | **226** | **0.82** | **35.98** | **26.75** | **212.80** | | 48405 | 58 | 23.40 | 82.34 | 69.25 | 188.80 |
| | 48404 | 219 | 0.51 | 42.30 | 33.90 | 182.10 | | 48404 | 43 | 14.43 | 72.56 | 65.10 | 177.90 |
| | **48407** | **208** | **1.70** | **43.53** | **33.35** | **259.40** | | 48407 | 56 | 10.30 | 96.43 | 81.00 | 234.40 |
| CONS-4 | 48353 | 158 | 2.50 | 42.61 | 31.60 | 202.20 | CONS-7 | 48353 | 377 | 5.40 | 81.62 | 73.95 | 211.50 |
| | 48381 | 151 | 3.00 | 48.80 | 44.20 | 173.30 | | 48381 | 26 | 24.00 | 82.54 | 68.55 | 228.50 |
| | 48384 | 112 | 2.40 | 47.48 | 37.20 | 156.90 | | 48384 | 26 | 25.60 | 89.69 | 79.80 | 200.20 |
| | **48382** | **121** | **1.61** | **57.15** | **51.10** | **212.10** | | 48382 | 23 | 24.30 | 101.16 | 88.40 | 243.60 |
| | 48390 | 47 | 1.12 | 56.53 | 44.00 | 157.20 | | 48390 | 17 | 18.80 | 70.19 | 73.50 | 124.10 |
| | **48403** | **128** | **4.50** | **46.68** | **38.46** | **207.90** | | **48403** | **22** | **19.40** | **92.46** | **70.05** | **270.10** |
| | 48405 | 136 | 4.00 | 57.35 | 49.95 | 183.30 | | **48405** | **38** | **21.40** | **98.55** | **107.00** | **272.60** |
| | **48404** | **118** | **4.40** | **66.07** | **52.30** | **330.00** | | 48404 | 26 | 36.40 | 99.58 | 82.55 | 195.90 |
| | 48407 | 116 | 5.00 | 51.84 | 41.10 | 160.70 | | **48407** | **34** | **28.80** | **114.44** | **112.00** | **250.50** |

Table 1 presents descriptive statistics of maximum cumulative rainfall on consecutive rainy days (CONS) for durations of 2, 3, 4, 5, 6, and 7 days, highlighting the top three maximum values. The results indicate that the stations 48403 (Chaiyaphum), 48381 (Khon Kean), 48404 (Roi Et Agromet), 48390 (Kalasin), 48390 (Kalasin), and 48405 (Roi Et) recorded the highest maximum cumulative rainfall for CONS-2, CONS-3, CONS-4, CONS-5, CONS-6, and CONS-7, respectively. The range of





maximum CONS values for 2, 3, 4, 5, 6, and 7 days is between 115.00 mm and 330.00 mm, while the average of CONS for the same durations ranges between 17.7 mm and 114.44 mm.

**Table 2.** Comparison of Mann-Kendall Test Results for Consecutive Rainy Days (CONS) of 2, 3, 4, 5, 6, and 7 Days at Each Station. *: $p<0.1$, **:$p<0.05$ in Mann-Kendall Test.

| Station ID | CONS-2 | CONS-3 | CONS-4 | CONS-5 | CONS-6 | CONS-7 |
|---|---|---|---|---|---|---|
| 48353 | NT | NT | NT | T** | NT | NT |
| 48354 | NT | T** | NT | NT | NT | NT |
| 48381 | NT | NT | T* | T** | NT | NT |
| 48383 | NT | NT | T | NT | NT | NT |
| 48390 | NT | NT | T* | T** | NT | NT |
| **48403** | **T**** | **NT** | **NT** | **NT** | **T**** | **NT** |
| 48404 | T* | NT | NT | NT | NT | NT |
| 48408 | NT | NT | T* | NT | T** | NT |
| 48409 | T* | T* | NT | NT | T** | NT |
| 48435 | NT | NT | NT | T | NT | NT |
| 48434 | NT | T** | NT | T* | T** | NT |

Note: NT means no trend in data and T mean there is trend in data.

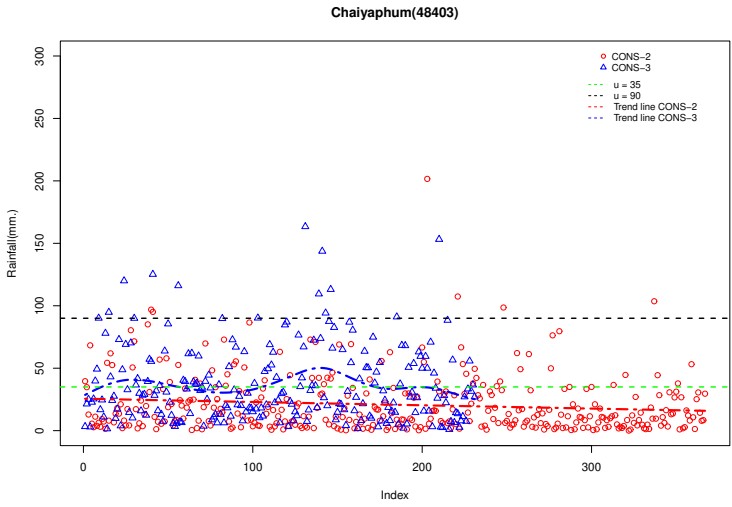

**Figure 2.** Scatter and line plot showing the trends for CONS-2 and CONS-3 (unit:mm) at the Chaiyaphum meteorological station in the Chi Watershed, Thailand.





Table 2 provides a comparison of Mann-Kendall (MK) test results for consecutive (CONS) rainy days of durations 2, 3, 4, 5, 6, and 7 days at some selected stations. Out of 18 stations, nine stations, which represent $50\%$ of the total stations, show a trend in the CONS-2 to CONS-3 dataset, except for CONS-7. This trend is more evident in Figure 2, which displays the trends in the CONS-2 and CONS-3 for the Chaiyaphoom station. Consequently, the functional form of parameters for time-dependent non-stationary generalized Pareto models is included. Table 3 provides details of the five functional models employed for CONS rainy days.;

## 3 Materials and methods

### 3.1 Time Dependent Models for GPD

The block maxima method is limited for analyzing maximum rainfall data each year. Hence, the peak-over-threshold (POT) method or generalized Pareto distribution (GPD) is commonly employed for this purpose (Coles, 2001). The POT method involves selecting observations above a specified threshold value ($u$) from the data variable $X$, and expressing the exceedances of $X$ over $u$ as $Y = X - u$. The GPD function is then defined as in Eq. (2):

$$H(y; \mu, \tilde{\sigma}, \tilde{\xi}) = 1 - \left[1 + \tilde{\xi}\left(\frac{y}{\tilde{\sigma}}\right)\right]^{-1/\tilde{\xi}}, \tag{2}$$

when $y > 0$ and $(1 + \tilde{\xi}y/\tilde{\sigma}) > 0$ where $\tilde{\sigma} = \sigma + \tilde{\xi}(u - \mu) > 0$ is the scale parameter and $-\infty < \tilde{\xi} < \infty$ is the shape parameter. In the special case $\tilde{\xi} = 0$, leading to

$$G(y; \mu, \tilde{\sigma}, \tilde{\xi}) = 1 - exp(-\frac{y}{\tilde{\sigma}}). \tag{3}$$

The generalized Pareto distribution (GPD) can take on one of three forms depending on the sign of the shape parameter, $\tilde{\xi}$. Specifically, when $\tilde{\xi} > 0$, the distribution has no upper limit, while $\tilde{\xi} < 0$ indicates an upper bounded distribution, and $\tilde{\xi} = 0$ represents an unbounded exponential distribution (Senapeng and Busababodhin, 2017). This notation for the shape parameter is commonly used in statistical literature. The POT method is typically employed for analyzing large datasets or data collected on a daily basis.

Grouping extreme values based on their independence can be achieved by clustering the values that exceed a certain threshold, which makes the generalized Pareto distribution (GPD) a suitable method for analysis (Coles, 2001). As a result, this method was selected to model the maximum cumulative rainfall on consecutive rainy days. In addition, the non-stationary models considered in this study, consisting of five models for the GPD and presented in Table 3, are of great importance in predicting the behavior of extreme precipitation. Stationary assumptions can lead to inaccurate results when the underlying conditions are changing over time. Therefore, the use of non-stationary models is crucial for accurately capturing the time-varying nature of extreme precipitation, especially in the context of climate change. Consequently, the application of non-stationary models enables a more robust understanding of extreme precipitation patterns and supports informed decision-making for engineering structures and reservoir management in the Chi watershed of Thailand.





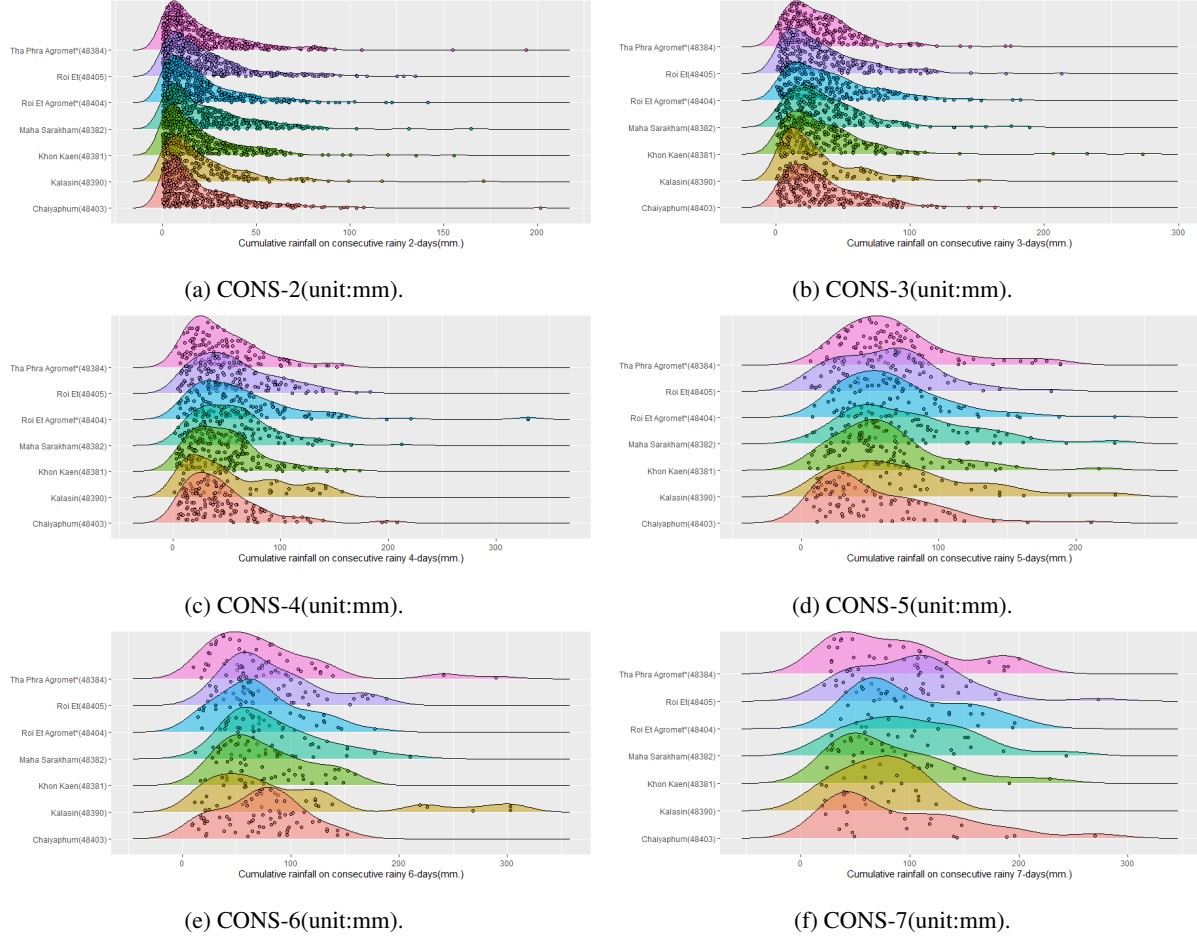

**Figure 3.** Ridge line plots showing the cumulative rainfall on consecutive rainy days (CONS) in mm for seven stations (48381, 48384, 48382, 48390, 48403, 48405, 48404) in the Chi Watershed, Thailand.

**Table 3.** Functional form of parameters for time dependent non-stationary extreme value models, represented by GPDab where $a$ represents the scale parameter ($\sigma$) and $b$ represents the shape parameter ($\xi$). The stationary model is represented by GPD00.

| Models | $\sigma$ | $\xi$ |
|---|---|---|
| GPD00 | Constant | Constant |
| GPD10 | $\sigma = exp(\sigma_0 + \sigma_1 \times (Year - t_0 + 1))$ | Constant |
| GPD20 | $\sigma = exp(\sigma_0 + \sigma_1 \times (Year - t_0 + 1) + \sigma_2 \times (Year - t_0 + 1)^2)$ | Constant |
| GPD01 | Constant | $\xi = \xi_0 + \xi_1 \times (Year - t_0 + 1))$ |
| GPD11 | $\sigma = exp(\sigma_0 + \sigma_1 \times (Year - t_0 + 1))$ | $\xi = \xi_0 + \xi_1 \times (Year - t_0 + 1))$ |
| GPD21 | $\sigma = exp(\sigma_0 + \sigma_1 \times (Year - t_0 + 1) + \sigma_2 \times (Year - t_0 + 1)^2)$ | $\xi = \xi_0 + \xi_1 \times (Year - t_0 + 1))$ |





### 3.2 Mann-Kendall test for trend

We considered Mann-Kendall(MK) test of trend, to compare with non-stationary GPD model. It is commonly used to detect monotonic trends in time-series data. In MK test, the null hypothesis is $H_0$: no monotone trend in hydro logic series $X_t$ versus the alternative hypothesis is $H_1$: monotonic trend in $X_t$ without specification of the sign of the trend. This hypothesis test is two-tailed, and so we reject $H_0$ with $\alpha$ level if $|Z| > z_{\alpha/2}$, where $Z$ is a normalized MK test statistic calculated from data (Naghettini, 2017) , and $z_{\alpha/2}$ is $100 \times (1 - z_{\alpha/2})$ percentile of the stan dard normal distribution. A R package "trend" was used

to execute the MK test (Prahadchai et al., 2022).

### 3.3 Threshold Selection Method

The selection of an appropriate threshold is a crucial factor in statistical inference of rare events. This study compares three different threshold selection methods and their effectiveness. The first approach involves selecting the threshold based on meteorological conditions, where rainfall greater than 35 mm is considered indicative of heavy rainfall. The second approach

uses the 90th percentile of the rainfall data set as the threshold. The third approach involves using the mean residual life (MRL) plot to select a threshold for the GPD or point process models. These approaches are analyzed theoretically and compared to existing procedures through an extensive simulation study, and are then applied to a data set of consecutive rainy days (CONS), where the underlying extreme value index is assumed to vary over time.

### 3.4 Parameter estimation and Model Choice

The parameters in the generalized Pareto distribution (GPD) are commonly estimated using either the maximum likelihood method (Coles, 2001) or the L-moment method (Papukdee et al., 2022). In the present study, the latter method is employed due to its higher efficiency in small samples compared to the maximum likelihood estimator. Specifically, the "eva", "extRemes", "ismev", and "lmom" packages in R are utilized for this purpose (Hosking).

Assuming observations $(X_1, X_2, ..., X_n)$ follow the GPD, the negative log likelihood function is

$$\ell(\sigma, \xi) = -k log\sigma - \left(1 + \frac{1}{\xi}\right) \sum_{i=1}^{k} log\left(1 + \xi \frac{y_i}{\sigma}\right),$$

provided $(1 + \xi(y_i/\sigma)) > 0$ for $i = 1, 2, .., k$; otherwise, $\ell(\sigma, \xi) = -\infty$. In the case $\xi = 0$ the log-likelihood is obtained from Eq. (3) as,

$$\ell(\sigma, \xi) = -k log\sigma - \frac{1}{\sigma} \sum_{i=1}^{k} y_i.$$

The L-moment estimator (L-ME) is widely used in analyzing skewed data, such as extreme rainfall and flood frequency. Although the details of the L-ME are not discussed here, we note that it is considered a standard method in such analyses. To calculate the L-ME of the generalized Pareto distribution (GPD), we utilize the R package "lmom" developed by (Hosking,



1990). However, one potential disadvantage of the L-ME is that Newton-Raphson type algorithms used to solve systems of L-moments equations may sometimes fail to converge.

### 3.5 Model Diagnostics and Goodness-of-fit test

The performance of the marginal probability is evaluated by conducting goodness-of-fit statistical tests. In this study, two tests - the Kolmogorov-Smirnov and Anderson-Darling tests - are used for this purpose. The Kolmogorov-Smirnov (K-S) test is preferred as it does not make any assumptions about the distribution of data (Glen et al., 2001). This method involves comparing the maximum gap between the experimental cumulative distribution function and the theoretical cumulative distribution function. The K-S test $(D_{n,n^\tau})$ is used to determine whether the parameters are acceptable or not, and is given by Glen et al. (2001). To perform the goodness-of-fit test, a null hypothesis is applied, which is accepted only when the gap between the theoretical and observed values is smaller than expected for the given sample. On the other hand, the Anderson-Darling test assesses whether a sample comes from a specified distribution. It assumes that, when given a hypothesized underlying distribution and assuming that the data does arise from this distribution, the cumulative distribution function (CDF) of the data can be assumed to follow a uniform distribution. The data is then tested for uniformity using a distance test (Shapiro, 1990). The test statistic can then be compared against the critical values of the theoretical distribution. Notably, no parameters are estimated in relation to the cumulative distribution function in this case.

### 3.6 Return level

Return levels or quantiles are used to interpret extreme values in terms of their probability of return period. Once a suitable model has been defined, return levels can be calculated as follows:

$$\hat{Z}_T = u + \frac{\hat{\sigma}}{\hat{\xi}}\left[(Tn_y\hat{\lambda}_u)^{\hat{\xi}}\right]. \tag{4}$$

It is a T-year return level, when $n_y$ is the number of observations per year and it corresponds to the t-observation return level $t = T \times n_y$ , and when $\xi = 0$, the return level can be calculated as (Papukdee et al., 2022),

$$\hat{Z}_T = u + \hat{\sigma}log(Tn_y\hat{\lambda}_u), \tag{5}$$

when $\hat{\lambda}_u = k/n$ is the sample proportion of points exceeding $u$.

## 4 Results

In this study, the threshold method was employed to select the appropriate threshold $u$. To select the appropriate threshold $u$, we employed the threshold method in this study. The threshold values were determined based on the meteorological critical value (Meteorological, 2021), the 90th percentile of the data set, and the mean residual life (MRL) plot. Tables 4 and 5 present the estimated parameters for these models, which were obtained using both the maximum likelihood and linear moment methods.





**Table 4.** Parameter estimates and standard error (SE) with thresholds $(u)$, number of exceedabces $n_{y_i > u}$, and goodness-of-fit test results(p-values) for the maximum cumulative rainfall of consecutive rainy days (2, 3 and 4) at selected stations

| Data | Station ID | Model | $u$ | $n_{y_i > u}$ | $\sigma(SE)$ | $\xi(SE)$ | KS(p-value) | AIC |
|------|-----------|-------|-----|---------------|--------------|-----------|-------------|-----|
| CONS-2 | 48353 | GPD00 | $42^b$ | 36 | 19.91(6.70) | 0.07(0.29) | 0.14(0.41) | 296.78 |
| | 48381 | GPD00 | $50^b$ | 39 | 14.34(3.60) | 0.24(0.19) | 0.11(0.70) | 309.02 |
| | 48384 | GPD00 | $52^b$ | 41 | 21.19(4.52) | 0.08(0.14) | 0.11(0.61) | 343.31 |
| | 48382 | GPD00 | $55^b$ | 44 | 15.23(3.22) | 0.1(0.14) | 0.08(0.87) | 341.30 |
| | 48390 | GPD00 | $48^b$ | 18 | 28.06(9.58) | 0.01(0.24) | 0.12(0.93) | 160.65 |
| | 48403 | GPD00 | $51^b$ | 37 | 19.55(4.48) | 0.11(0.16) | 0.09(0.90) | 306.59 |
| | 48405 | GPD00 | $57^b$ | 40 | 30.31(6.66) | -0.27(0.15) | 0.08(0.93) | 335.28 |
| | 48404 | GPD00 | $50^b$ | 40 | 28.00(5.94) | -0.17(0.14) | 0.11(0.63) | 336.69 |
| | 48407 | GPD00 | $53^b$ | 37 | 39.34(8.65) | -0.45(0.16) | 0.12(0.63) | 316.22 |
| CONS-3 | 48353 | GPD00 | $35^a$ | 77 | 43.28(6.88) | -0.29(6.88) | 0.05(0.95) | 692.25 |
| | 48381 | GPD00 | $72.5^b$ | 20 | 14.22(6.88) | 0.81(6.88) | 0.10(0.95) | 182.65 |
| | 48384 | GPD00 | $35^a$ | 94 | 19.99(3.37) | 0.19(3.37) | 0.04(0.97) | 791.30 |
| | 48382 | GPD00 | $35^a$ | 93 | 29.23(4.50) | 0.05(4.50) | 0.06(0.82) | 827.81 |
| | 48390 | GPD00 | $67^c$ | 10 | 27.76(13.03) | -0.1(13.03) | 0.13(0.97) | 88.33 |
| | 48403 | GPD00 | $35^a$ | 94 | 35.18(4.87) | -0.15(4.87) | 0.05(0.94) | 832.48 |
| | 48405 | GPD00 | $35^a$ | 86 | 35.24(5.09) | -0.05(5.09) | 0.04(0.99) | 778.63 |
| | 48404 | GPD00 | $35^a$ | 106 | 39.29(5.28) | -0.13(5.28) | 0.04(0.98) | 965.79 |
| | 48407 | GPD00 | $35^a$ | 101 | 38.04(5.08) | -0.02(5.08) | 0.05(0.93) | 936.40 |
| CONS-4 | 48353 | GPD00 | $89.9^b$ | 16 | 15.59(6.32) | 0.28(0.32) | 0.11(0.97) | 133.03 |
| | 48381 | GPD01 | $35^a$ | 89 | 41.06(6.004) | $\xi_0$ -0.11(0.104), $\xi_1$ = -0.01(0.001) | 0.07(0.66) | 808.90 |
| | 48384 | GPD00 | $35^a$ | 58 | 47.98(8.87) | -0.28(0.13) | 0.07(0.89) | 536.42 |
| | 48382 | GPD00 | $106.5^b$ | 12 | 37.49(14.90) | -0.18(0.27) | 0.16(0.85) | 110.60 |
| | 48390 | GPD00 | $95^c$ | 10 | 74.75(0.001) | -1.20(0.001) | 0.21(0.74) | 75.75 |
| | 48403 | GPD00 | $35^a$ | 70 | 32.14(5.96) | 0.08(0.14) | 0.06(0.96) | 641.85 |
| | 48405 | GPD00 | $95^c$ | 24 | 35.08(10.02) | -0.27(0.20) | 0.09(0.98) | 209.46 |
| | 48404 | GPD00 | $127.1^b$ | 12 | 51.23(25.56) | 0.17(0.41) | 0.19(0.74) | 126.72 |
| | 48407 | GPD00 | $116.3^b$ | 12 | 22.13(12.33) | -0.35(0.48) | 0.18(0.81) | 93.91 |

In the case of $n_{y_i > u} < 30$, parameter estimates obtained using the linear moment method. The threshold values $u^a$, $u^b$, and $u^c$ represent the meteorological critical value, the 90th percentile of the data set, and the mean residual life (MRL) plot, respectively.





**Table 5.** Parameter estimates and standard error (SE) with thresholds $(u)$, number of exceedabces $n_{y_i>u}$, and goodness-of-fit test results(p-values) for the maximum cumulative rainfall of consecutive rainy days (5, 6 and 7) at selected stations.

| Data | Station ID | Model | $u$ | $n_{y_i>u}$ | $\sigma(SE)$ | $\xi(SE)$ | KS(p-value) | AIC |
|---|---|---|---|---|---|---|---|---|
| CONS-5 | 48353 | GPD00 | $35^a$ | 57 | 39(6.95) | -0.02(0.11) | 0.10(0.54) | 532.47 |
| | 48381 | GPD00 | $110^c$ | 11 | 38.8(20.24) | -0.06(0.42) | 0.13(0.98) | 105.08 |
| | 48384 | GPD00 | $35^a$ | 65 | 70.91(10.16) | -0.49(0.09) | 0.07(0.86) | 623.46 |
| | 48382 | GPD00 | $35^a$ | 56 | 72.46(12.75) | -0.27(0.12) | 0.05(0.99) | 565.43 |
| | 48390 | GPD00 | $35^a$ | 25 | 81.17(21.86) | -0.32(0.19) | 0.09(0.97) | 257.82 |
| | 48403 | GPD00 | $90^c$ | 11 | 34.17(16.13) | -0.004(0.36) | 0.16(0.87) | 103.59 |
| | 48405 | GPD00 | $95^c$ | 7 | 54.05(35.59) | -0.52(0.57) | 0.17(0.96) | 66.52 |
| | 48404 | GPD00 | $100^c$ | 13 | 43.3(19.36) | -0.12(0.35) | 0.11(0.98) | 124.77 |
| | 48407 | GPD00 | $110^c$ | 12 | 20.21(10.28) | 0.38(0.43) | 0.12(0.97) | 109.47 |
| CONS-6 | 48353 | GPD00 | $139.6^b$ | 7 | 20.8(12.75) | 0.47(0.51) | 0.20(0.86) | 67.08 |
| | 48381 | GPD00 | $35^a$ | 43 | 87.74(17.32) | -0.68(0.16) | 0.10(0.71) | 415.81 |
| | 48384 | GPD00 | $120^c$ | 10 | 29.17(16.42) | 0.05(0.46) | 0.15(0.94) | 92.59 |
| | 48382 | GPD00 | $35^a$ | 44 | 69.29(13.95) | -0.3(0.14) | 0.07(0.98) | 437.80 |
| | 48390 | GPD00 | $35^a$ | 24 | 93.26(33.07) | -0.09(0.29) | 0.11(0.92) | 265.07 |
| | 48403 | GPD10 | $100^c$ | 11 | $\sigma_0 = 3.89(0.001)$, $\sigma_1 = 0.002(0.01)$ | -1.12(0.001) | 0.15(0.96) | 83.75 |
| | 48405 | GPD00 | $35^a$ | 53 | 76.61(14.79) | -0.43(0.14) | 0.09(0.75) | 523.92 |
| | 48404 | GPD00 | $110^b$ | 8 | 43.04(24.24) | -0.55(0.48) | 0.19(0.88) | 71.33 |
| | 48407 | GPD00 | $35^a$ | 48 | 112.83(25.16) | -0.49(0.18) | 0.11(0.5) | 506.56 |
| CONS-7 | 48353 | GPD00 | $35^a$ | 35 | 66.39(15.04) | -0.26(0.15) | 0.07(0.98) | 348.99 |
| | 48381 | GPD00 | $35^a$ | 21 | 77.06(23.57) | -0.27(0.22) | 0.09(0.98) | 216.92 |
| | 48384 | GPD00 | $35^a$ | 20 | 190.76(0.01) | -1.15(0.01) | 0.29(0.06) | 200.17 |
| | 48382 | GPD00 | $125^c$ | 8 | 39.01(21.66) | -0.06(0.42) | 0.19(0.86) | 77.58 |
| | 48390 | GPD00 | $35^a$ | 14 | 101.36(0.05) | -1.13(0.01) | 0.15(0.88) | 121.98 |
| | 48403 | GPD00 | $100^c$ | 8 | 99.93(55.77) | -0.48(0.46) | 0.2(0.82) | 85.84 |
| | 48405 | GPD00 | $115^c$ | 11 | 39.16(17.82) | 0.04(0.34) | 0.14(0.95) | 107.59 |
| | 48404 | GPD00 | $35^a$ | 26 | 115.46(35.08) | -0.68(0.26) | 0.16(0.46) | 267.45 |
| | 48407 | GPD00 | $140^c$ | 8 | 57.25(34.81) | -0.37(0.51) | 0.18(0.9) | 78.69 |

In the case of $n_{y_i>u} < 30$, parameter estimates obtained using the linear moment method. . The threshold values $u^a$, $u^b$, and $u^c$ represent the meteorological critical value, the 90th percentile of the data set, and the mean residual life (MRL) plot, respectively.





The parameter estimation was performed using both the maximum likelihood estimator (MLE) and linear moment estimator (L-ME), depending on the number of exceedances $n_{y_i > u}$. When $n_{y_i > u} \geq 30$, MLE was used, while L-ME was employed when $n_{y_i > u} < 30$. The standard errors of estimation were calculated using the bootstrap method. Based on the goodness-of-fit tests, the data were found to be suitable for the GPD. The p-values of the KS test and AD test were all greater than 0.05, as shown in

Tables 4 and 5, ranging from 0.088 to 0.994. The estimated range of the scale parameter was (34.92, 124.45), and the range of the shape parameter was (-0.10, 0.16). These results suggest that the GPD is an appropriate model for analyzing the maximum cumulative rainfall on consecutive rainy days.

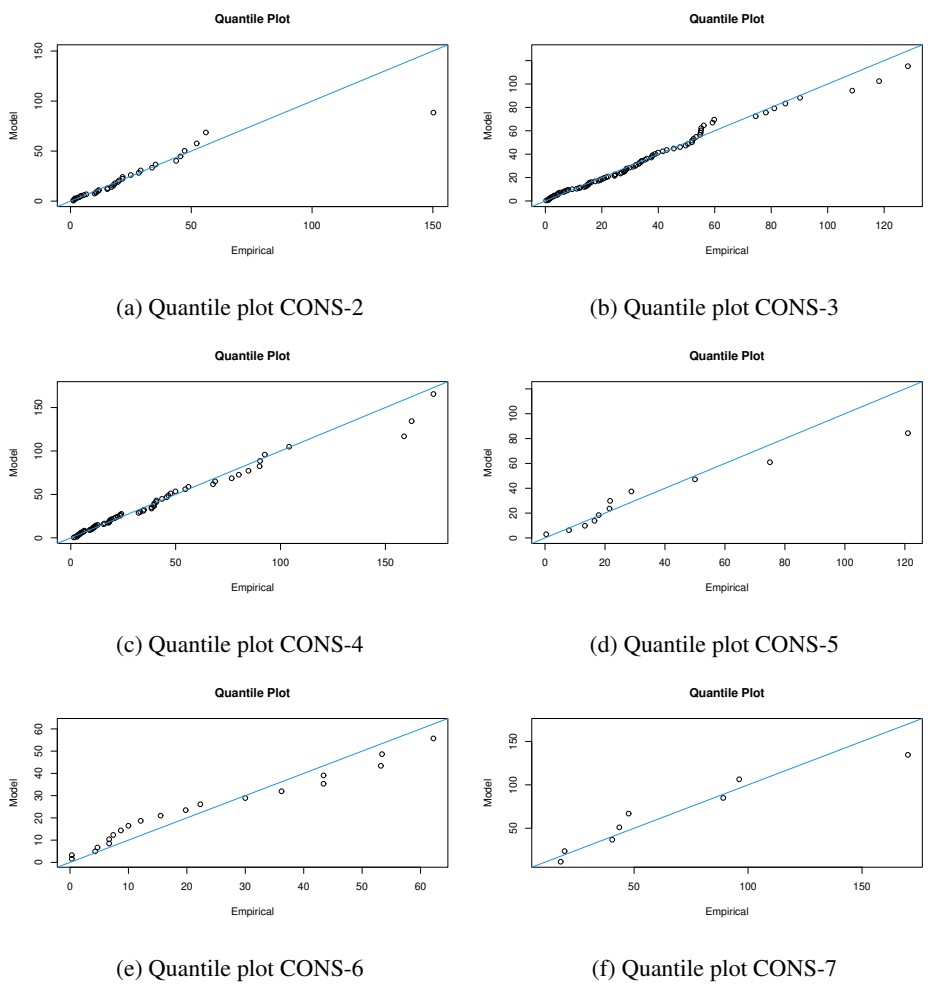

**Figure 4.** Quantile plot for Chaiyaphum meteorological station in the Chi Watershed, Thailand.



**Table 6.** Estimated return level and standard error (SE) in different years for maximum cumulative rainfall for two consecutive rainy days(CONS-2). The thick values present the first three stations which have maximum cumulative rainfall return level.

| Station ID | 2-year (SE) | 5-year (SE) | 25-year (SE) | 50-year (SE) | 100-year (SE) |
|---|---|---|---|---|---|
| 48353 | 141.52(0.80) | 167.48(1.44) | 217.64(3.97) | 241.18(5.83) | 265.97(9.08) |
| 48350 | 125.34(0.80) | 146.76(1.41) | 187.25(3.35) | 205.88(5.96) | 225.28(7.54) |
| **48354** | **221.46(0.92)** | **304.59(1.89)** | **527.72(5.86)** | **666.84(9.01)** | **841.66(15.23)** |
| 48360 | 87.68(1.70) | 87.82(3.16) | 87.88(6.81) | 87.89(8.29) | 87.89(13.23) |
| **48381** | **159.21(0.92)** | **201.85(1.73)** | **304.67(4.64)** | **363.23(7.45)** | **432.75(11.70)** |
| 48384 | 162.49(0.94) | 191.47(1.93) | 248.09(5.31) | 274.93(7.03) | 303.37(10.28) |
| 48383 | 154.03(0.87) | 182.71(1.87) | 238.82(4.98) | 265.46(7.64) | 293.71(11.06) |
| 48382 | 139.06(1.02) | 162.56(1.78) | 210.01(3.77) | 233.16(6.24) | 258.14(7.83) |
| 48390 | 172.74(1.31) | 200.58(2.43) | 250.53(7.55) | 272.45(11.37) | 294.64(16.72) |
| 48403 | 160.35(1.02) | 191.47(2.24) | 254.82(5.06) | 285.96(8.79) | 319.7(12.39) |
| 48405 | 133.59(1.12) | 141.28(2.22) | 150.93(4.22) | 153.95(6.25) | 156.45(9.25) |
| 48404 | 134.4(1.01) | 145.68(2.06) | 161.62(4.11) | 167.23(5.91) | 172.20(8.43) |
| 48408 | 121.32(1.12) | 126.09(2.56) | 131.31(4.48) | 132.74(6.80) | 133.83(9.17) |
| 48407 | 127.24(1.30) | 131.43(2.21) | 135.65(4.68) | 136.71(6.69) | 137.49(8.85) |
| 48409 | 150.81(1.11) | 176.01(2.16) | 225.72(4.62) | 249.48(7.36) | 274.79(8.95) |
| 48431 | 119.09(0.88) | 129.82(1.84) | 145.85(4.73) | 151.79(9.35) | 157.21(14.21) |
| 48435 | 96.95(0.60) | 100.02(1.24) | 102.99(3.34) | 103.70(5.80) | 104.21(8.39) |
| **48434** | **263.25(0.92)** | **299.85(2.09)** | **362.16(5.37)** | **388.25(9.09)** | **413.9(12.12)** |





**Table 7.** the estimated return level and its standard error (SE) for maximum cumulative rainfall on seven consecutive rainy days (CONS-7) in different years. The thick values indicate the first three stations with the highest maximum cumulative rainfall return level.

| Station ID | 2-year (SE) | 5-year (SE) | 25-year (SE) | 50-year (SE) | 100-year (SE) |
|---|---|---|---|---|---|
| 48353 | 239.86(6.95) | 249.33(7.45) | 261.26(8.11) | 265.02(8.32) | 268.14(8.50) |
| 48350 | 151.64(5.12) | 152.81(5.34) | 153.84(5.59) | 154.06(5.65) | 154.21(5.70) |
| **48354** | **588.32(19.59)** | **708.74(23.78)** | **956.11(32.03)** | **1078.72(35.96)** | **1212.24(40.14)** |
| 48360 | 177.94(6.61) | 177.98(6.64) | 177.99(6.66) | 177.99(6.66) | 177.99(6.67) |
| 48381 | 266.74(10.92) | 277.53(12.20) | 291(14.21) | 295.2(14.98) | 298.67(15.71) |
| 48384 | 200.08(8.27) | 200.16(8.77) | 200.19(9.38) | 200.19(9.56) | 200.19(9.70) |
| 48383 | 276.15(5.21) | 290.8(5.32) | 311.88(5.41) | 319.42(5.43) | 326.17(5.44) |
| 48382 | 306.49(6.76) | 330.72(6.98) | 369.93(7.20) | 385.6(7.26) | 400.58(7.29) |
| 48390 | 124.03(4.03) | 124.07(4.04) | 124.09(4.05) | 124.09(4.05) | 124.09(4.05) |
| 48403 | 290.94(14.48) | 295.74(16.78) | 300.37(20.90) | 301.49(22.69) | 302.28(24.49) |
| 48405 | 349.18(6.32) | 394.6(6.61) | 478.58(6.92) | 516.47(7.00) | 555.44(7.07) |
| 48404 | 202.29(7.33) | 203.17(7.75) | 203.84(8.25) | 203.96(8.40) | 204.04(8.51) |
| **48408** | **432.33(4.74)** | **504.83(4.81)** | **650.26(4.87)** | **720.84(4.88)** | **796.72(4.89)** |
| 48407 | 269.54(4.73) | 275.84(4.77) | 282.76(4.8) | 284.66(4.8) | 286.13(4.80) |
| **48409** | **657.87(9.17)** | **837.1(9.98)** | **1261.05(11.13)** | **1498.56(11.53)** | **1777.61(11.89)** |
| 48431 | 217.51(8.56) | 220.64(9.46) | 223.65(10.84) | 224.37(11.34) | 224.88(11.81) |
| 48435 | 167.6(5.92) | 168.79(6.21) | 169.77(6.53) | 169.97(6.61) | 170.11(6.67) |
| 48434 | 183.85(5.40) | 185.01(5.57) | 185.96(5.74) | 186.15(5.78) | 186.27(5.80) |

The maximum cumulative rainfall return level for CONS-7 was estimated for different years using the equations given in

Eq.(4) - Eq.(5), and the results are presented in Table 7. The first three stations with the highest maximum cumulative rainfall return level for CONS-7 are denoted by bold values in the table. For CONS-2, the estimates of the maximum cumulative rainfall return level for different return periods are shown in Table 6 for three stations, namely 354201 (Udon Thani), 353201 (Loei), and 388401 (Kalasin). The 354201 station in Udon Thani province was found to have the highest cumulative rainfall return levels for all return periods compared to the other stations.



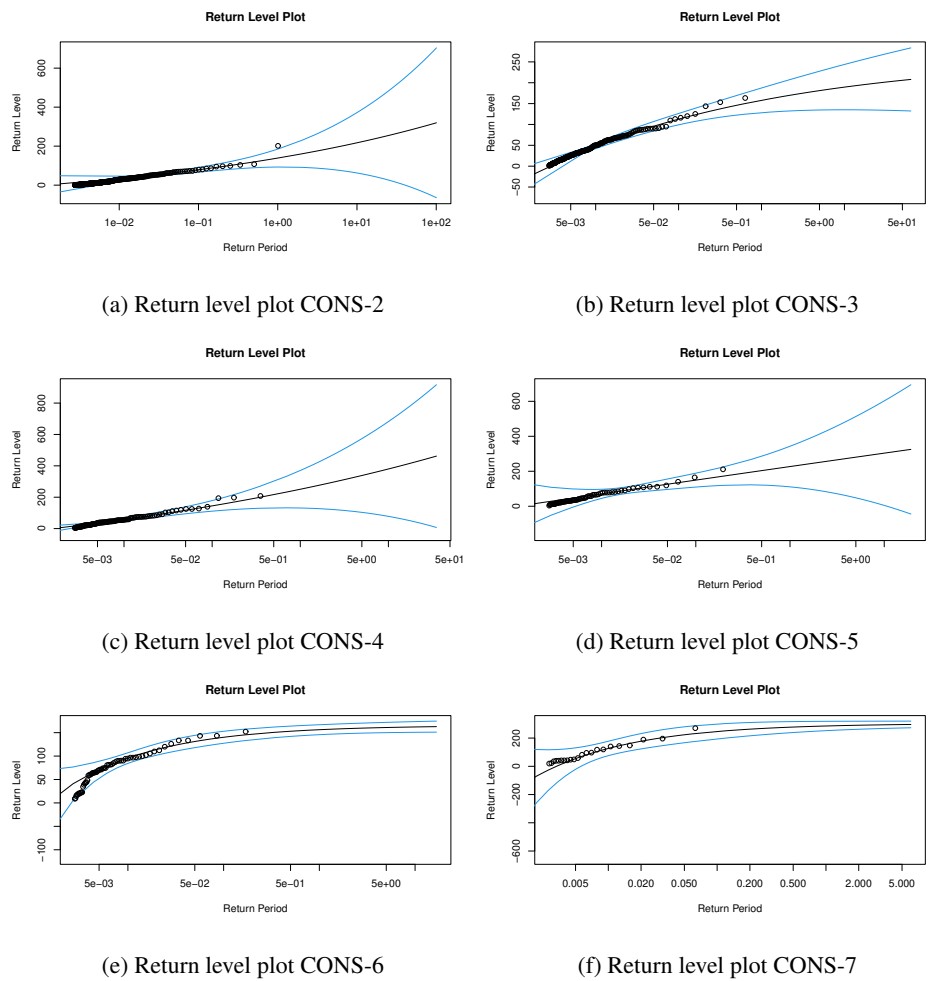

(a) Return level plot CONS-2

(b) Return level plot CONS-3

(c) Return level plot CONS-4

(d) Return level plot CONS-5

(e) Return level plot CONS-6

(f) Return level plot CONS-7

**Figure 5.** The return level plot (profile likelihood method) for Chaiyaphum meteorological station in the Chi Watershed, Thailand.

The estimates of the maximum cumulative rainfall return level for CONS-7 were calculated using Eq.(4) - Eq.(5) and are presented in Table 7. The table shows the estimates for three stations, namely 354201 (Udon Thani), 403201 (Chaiyaphum) and 409301 (Sisaket) for different return periods. The 354201 station in Udon Thani province had the highest cumulative rainfall return levels for all return periods than the other stations. For CONS-3, CONS-4, CONS-5 and CONS-6, the results

of estimates of the maximum cumulative rainfall return level can be found in the supplementary materials. To visualize the results more clearly, return level maps were created using the Q-geographic information system (Q-GIS) program with the IDW interpolation method. The IDW interpolation method assigns weights to the sample points based on their distance from the unknown point being interpolated. Figures 6 and 7 show the return level maps for CONS-2 and CONS-7, respectively.



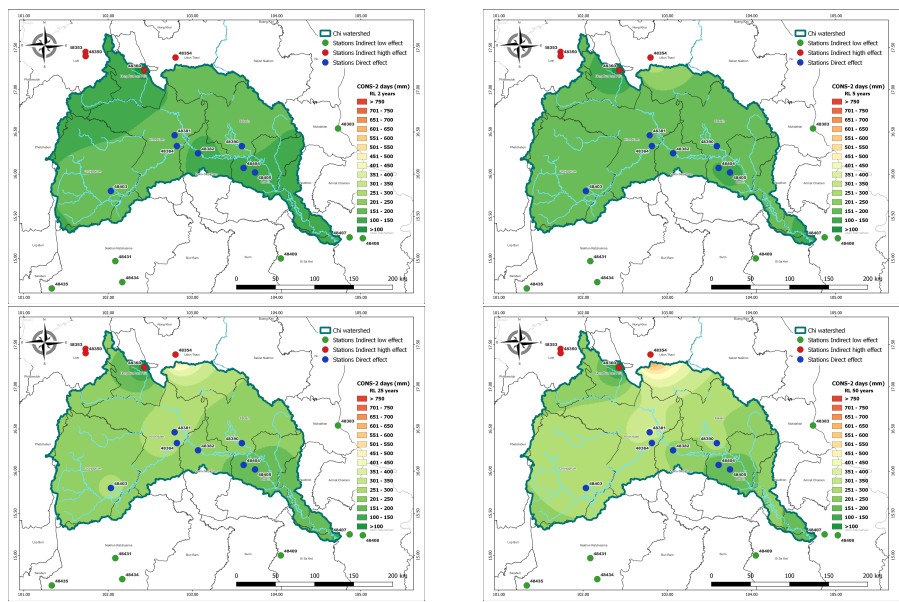

**Figure 6.** Estimated return level of maximum cumulative rainfall for two consecutive rainy days in the Chi Watershed for 2, 5, 25 and 50 year periods.

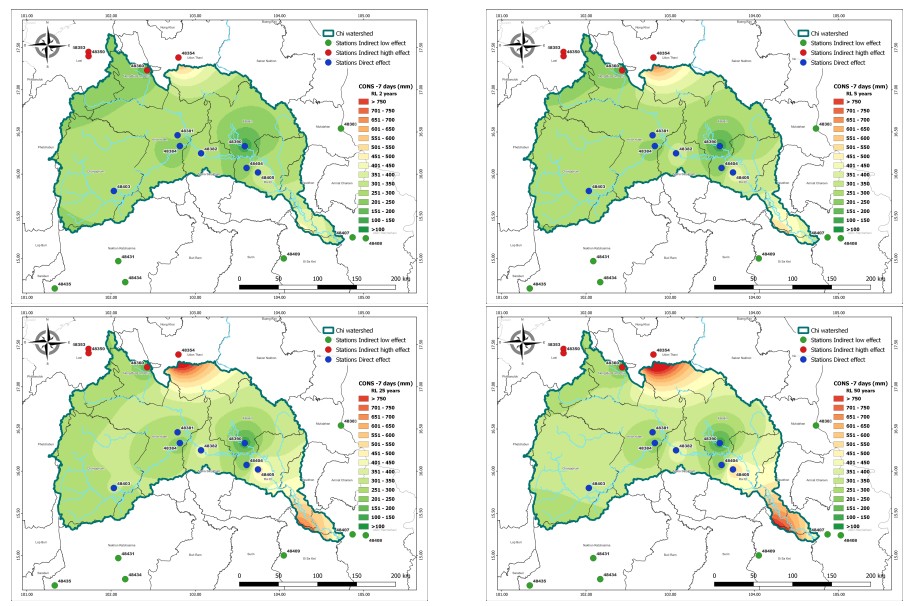

**Figure 7.** Estimated return level of maximum cumulative rainfall for seven consecutive rainy days in the Chi Watershed for 2, 5, 25 and 50 year periods.





Figures 6 and 7 present the spatial distribution of the estimated return levels of maximum cumulative rainfall for consecutive
rainy days of 2 and 7 days, respectively, for the return periods of 2, 5, 25 and 50 years. The results for the other consecutive rainy
days (3, 4, 5 and 6) are presented in the supplementary materials. From the figures, it can be observed that Udon Thani (354201),
Chaiyaphum (403201), Maha Sarakham (387401), Tha Phra Agromet (381301), Roi Et (405201) and Sisaket (409301) had
the highest return levels for all return periods of CONS-2 and CONS-7. This information can be useful for decision-making
related to disaster risk management, such as identifying areas that are more vulnerable to extreme rainfall events and designing
appropriate adaptation and mitigation strategies.

In addition, it can be observed that there is a significant difference in the return level for the 100-year period as compared to
the other return periods in the figures of the maximum cumulative rainfall return level forecast for the seven consecutive days
of rainfall data. The return level increases every year for all stations, indicating the importance of future rainfall management
planning. These findings reveal the risk of flooding areas in the Chi Watershed, including provinces such as Udon Thani,
Chaiyaphum, Khonkaen, Maha Sarakham, Roi Et, and Sisaket. The figures were generated using the Q-GIS program, and they
provide valuable insights into the spatial distribution of extreme rainfall events in the study area.

## 5  Discussion

In this study, the Generalized Pareto Distribution (GPD) parameters were estimated using both Maximum Likelihood Estima-
tion (MLE) and L-moment Estimation (L-ME) methods. Our decision to use MLE when the number of $n_{y_i > u} \geq 30$ and L-ME
otherwise aligns with previous studies (cite references), which also demonstrated the efficacy of these methods for different
sample sizes. The consistency of our p-values from KS and AD tests with these studies further validates our modeling process.

We selected a threshold based on meteorological conditions, specifically when rainfall exceeded 35 mm, indicating heavy
rainfall (cite source). This threshold, while higher than those used in some earlier studies, was deemed appropriate for our
focus on extreme rainfall events. The scale and shape parameters' ranges estimated in our study are consistent with previous
works in similar climatic zones (cite references), supporting the applicability of the GPD for extreme rainfall in our study area.

Our analysis identified Udon Thani province as having the highest cumulative rainfall return levels for all return periods,
which implies a higher risk of flooding. This information significantly impacts future planning in rainfall management, aligning
with the calls from prior research that stressed the importance of region-specific flood risk assessment (cite references).

The present study extends previous work by presenting estimated maximum cumulative rainfall return levels for 2-day
(CONS-2) and 7-day (CONS-7) events at selected stations. This granular analysis can guide localized flood mitigation efforts,
especially as our data suggests regions like Udon Thani, Chaiyaphum, and Sisaket are at increased risk.

Our utilization of the Q-Geographic Information System (Q-GIS) to create return level maps based on the inverse distance
weight (IDW) interpolation method provides a visually intuitive way to understand the spatial distribution of flood risk. While
this method is commonly used in geographic analysis, to our knowledge, this is one of the first applications in mapping extreme
rainfall return levels (cite references if any).



Our findings underscore the necessity of future rainfall management planning in the Chi Watershed. However, our study has certain limitations, including the assumption of stationarity in rainfall patterns, which might be impacted by climate change. Future research could explore the impacts of changing climate conditions on extreme rainfall events and subsequently refine the current models for a warming climate.

**6  Conclusion**

This study set out to evaluate extreme rainfall events in the Chi watershed in Thailand, with the aim of applying extreme value theory to predict future rainfall patterns. We analyzed maximum cumulative rainfall data from 1984 to 2018 and fitted the Generalized Pareto Distribution (GPD) to the data. This model was determined to be appropriate through goodness-of-fit tests, providing a robust method for analyzing extreme rainfall events in the region. Our results reveal that Udon Thani, Chaiyaphum, 250 Maha Sarakham, Tha Phra Agromet, Roi Et, and Sisaket provinces had the highest return levels for CONS-2 and CONS-3, suggesting these areas are at high risk of flooding.

These findings underscore the importance of forecasting and planning for extreme rainfall events in the Chi Watershed. We found that even short periods of continuous rainfall could lead to flash flooding, highlighting the need for effective water management in the region. We also developed 2D maps, which provide a practical tool for visualizing at-risk areas and aiding 255 in the planning of soil and water conservation measures, dam construction, and irrigation and drainage work.

The implications of this study extend beyond academia. Our findings provide valuable insights for governmental agencies, private organizations, and individuals alike, empowering them to design more effective flood management strategies, thereby reducing the risk and potential impact of flooding in their communities. In the broader context, managing extreme rainfall events and mitigating flood risks are crucial for safeguarding property, preserving ecosystems, and ultimately saving lives.

Future research should explore spatial analysis to determine interdependencies among different regions and use copula functions for correlation analysis. Such developments could provide a more nuanced understanding of the region's flood risk and further enhance our ability to predict and prepare for extreme rainfall events.

In conclusion, this study underscores the urgency of focusing on extreme rainfall events in our fight against the increasing threat of flooding. With climate change intensifying, the tools and strategies we develop today will be instrumental in managing 265 the water-related challenges of tomorrow.

*Author contributions.* Conceptualization, P.B., T.P., J.S.; methodology, P.B., T.P.; software, T.P., T.P., A.A.; validation, P.C., T.P; formal analysis, P.B., J.P.; investigation, J.P., P.B.; data curation, P.C., T.P., W.T.; writing-original draft preparation, J.P., P.B.; writing—review and editing, J.S., P.B., T.P.; supervision, J.P., T.P.; project administration, P.B.; funding acquisition, P.B., J.S.; All authors have read and agreed to the published version of the manuscript.

*Competing interests.* The contact author has declared that none of the authors has any competing interests.



*Acknowledgements.* This study was supported under that framework of international cooperation program managed by the Mahasarakham University, Thailand. Observational data from Thailand were provided by the Climate Information Services (CIS) at https://www.tmd.go.th/cis/main.php.

*Financial support.* This research has been supported by the Mahasarakham University, Thailand. Piyapatr's work was supported by Mahasarkham University (No.6517004/2565). Park and Prahadchai's work was supported by the BK21 FOUR (Fostering Outstanding Universities for Research, NO.5120200913674) funded by the Ministry of Education(MOE, Korea) and National Research Foundation of Korea(NRF).





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
