# Peer review of "Employing the Generalized Pareto Distribution to Analyze Extreme Rainfall Events on Consecutive Rainy Days in Thailand's Chi Watershed: Implications for Flood Management"

_Hydrology and Earth System Sciences, 2023_

## Author Comment (AC2)

Dear Reviewer,

Thank you for taking the time to review our manuscript entitled "Employing the Generalized Pareto Distribution to Analyze Extreme Rainfall Events on Consecutive Rainy Days in Thailand's Chi Watershed: Implications for Flood Management". We sincerely appreciate your insightful feedback and are pleased to hear that you found the paper of interesting.

It is encouraging to learn that our derivations are regarded as elegant and correct, and that our explanations, while different from what you might personally prefer, are still deemed appropriate. We have always strived to present our findings in the clearest manner possible while retaining our unique perspective, and your comments reassure us that we've managed to strike a balance.

Your concise summarization of our study, highlighting the practical application of our theoretical results, particularly in the context of flood management and climate change, is very much appreciated. We concur with your sentiment on the significance of preparing for extreme rainfall events, and we're glad that our work resonates with this perspective.

Your recommendation for the acceptance of our manuscript in the journal Hydrology and Earth System Sciences is both encouraging and gratifying. We understand the importance of the review process, and we're thankful for your positive feedback and constructive input.

Once again, thank you for your time and consideration. We look forward to any further instructions or suggestions that may come as we progress through the publication process.

Warm regards,

Piyapatr Busababodhin and Co-authors

---

## Author Response (AR1)

**November 14, 2023**

**RE:  Reminder hess-2023-167 (author) - manuscript needs Revisions**

Dear Editor,

Thanks for your e-mail of November 10, 2023, which provided useful comments from the reviewers on our manuscript entitled "Employing the Generalized Pareto Distribution to Analyze Extreme Rainfall Events on Consecutive Rainy Days in Thailand's Chi Watershed: Implications for Flood Management". We have studied these comments carefully and have revised the manuscript according to most of the suggestions.

We would like to thank the reviewers for a careful and critical review of the paper.  We would also like to thank you for processing our manuscript.

Sincerely,
Piyapatr Busababodhin

Reviewers: Red highlights

**Title:** Employing the Generalized Pareto Distribution to Analyze Extreme Rainfall Events on Consecutive Rainy Days in Thailand's Chi Watershed: Implications for Flood Management

In their study, the authors have put forth the application of the Generalized Pareto Distribution (GPD) as a means to characterize the extreme rainfall data. They have estimated the distribution's parameters through both maximum likelihood and linear moment estimation methods. However, it is worth noting that relying solely on the GPD may be insufficient. It would be beneficial to incorporate additional distributions for comparative analysis. Furthermore, enhancing the analysis by taking into account spatial and temporal dependencies within the model could prove valuable.

| No | Comments | Details |
|---|---|---|
| 1. | Regarding the selection of the threshold "u," is it possible to treat it as a model parameter and estimate it directly from the data? | Thank you for comments, the $u$ value is used for the GP. This value is set to a constant value and the observed data with values greater than $u$ (that is, $X > u$). Thus, the cumulative distribution function of GPD is $\Pr(Y = X > u \mid Y \leq y)$.

Therefore, the value of u cannot be considered as another parameter in this distribution. But it is treated as the mean of the raw observed data when estimating the return level.

The value of $u$ must be large enough to obtain the maximum observed data and have an appropriate sample size.

It is possible to treat u as a model parameter and estimate it directly from the data. We will consider about your recommendation in our future work. |
| 2. | There appears to be some confusion regarding the notations used in equation (2) and in the negative log likelihood function, particularly on line 150 of page 9, and in several other instances. | Thank you for comments, we have corrected in equation 2.

$$H(y) = 1 - \left(1 + \frac{\xi y}{\tilde{\sigma}}\right)^{-\frac{1}{\xi}},$$
defined on $y > 0$, where $\tilde{\sigma} = \sigma + \xi(u - \mu)$ is the scale parameter and $-\infty < \xi < \infty$ is the shape parameter. In the case $\xi = 0$, leading to

$$H(y) = 1 - exp\left(-\frac{y}{\tilde{\sigma}}\right), y > 0$$

And check in line 150 of page 9 as recommended. |
| 3. | How can you ensure the convergence of estimates when using Maximum Likelihood Estimation (MLE)? Have you | Thank you for comments, we have implemented the suggested modifications outlined on page 10, specifically within the range of lines |

| No | Comments | Details |
|---|---|---|
| | experimented with different initial values in the R functions employed for estimation? | 158-159, through the addition of references. |
| | | How can you ensure the convergence of estimates when using Maximum Likelihood Estimation (MLE)? There are some references such as Dupuis and Winchester, 2001; Papukdee et al., 2022., presented one potential disadvantage of the L-ME is that Newton-Raphson type algorithms used to solve systems of L-moments equations may sometimes fail to converge. And one more reason is about sample size which the L-ME is employed due to its higher efficiency in small samples compared to the MLE (Hosking, 1990).

Have you experimented with different initial values in the R functions employed for estimation?
Reply: We experimented with Nelder-Mead method in the "ismev" package in R program |
| 4. | While the authors have conducted model diagnostics and assessed goodness-of-fit, it would be advisable to include information criteria such as AIC and BIC to facilitate the selection of a more suitable model. Additionally, providing insights into prediction accuracy for each model would be beneficial. | Thank you for comments. We give more detail of step of model conduction by using diagnostics and assessed good-of-fit. In addition, we have incorporated the BIC results into Tables 4 and 5 and has provided additional explanations in accordance with the recommendations. |
| 5. | How have you verified the stationarity of the data, and what measures are taken if it is found to be non-stationary?

Have you considered incorporating external variables, such as temperature or others, into the model to enhance the accuracy of the analysis? | Thank you for comments, we performed data analysis using the Mann-Kendall (MK) test, as displayed in Table 2. For non-stationary case, we applied five models which effected to each parameter, showed in Table 3.

In recently research, we analysis focused only on a single variable, namely, time. However, in the future, the we intends to develop models and explore additional variables. |

---

## Author Response (AR2)

**November 16, 2023**

**RE:  Reminder hess-2023-167 (author) - manuscript needs Revisions**

Dear Editor,

Thanks for your e-mail on November 16, 2023, which provided useful comments from the reviewers on our manuscript entitled "Employing the Generalized Pareto Distribution to Analyze Extreme Rainfall Events on Consecutive Rainy Days in Thailand's Chi Watershed: Implications for Flood Management". We corrected this comment carefully and revised the manuscript according to your suggestions.

We would like to thank the reviewers for a careful and critical review of the paper.  We would also like to thank you for processing our manuscript.

Sincerely,
Piyapatr Busababodhin

**Title:** Employing the Generalized Pareto Distribution to Analyze Extreme Rainfall Events on Consecutive Rainy Days in Thailand's Chi Watershed: Implications for Flood Management

| No | Comments | Details |
|---|---|---|
| 1. | With the next file upload request, please check your 'Author contribution': JS is not part of the author list (Do you mean JP?). | Thank you for comments.

 I corrected in the 'Author contribution' section I used "JP" instead of "JS."

 I appreciate your prompt attention to this matter. |

**Title:** Employing the Generalized Pareto Distribution to Analyze Extreme Rainfall Events on Consecutive Rainy Days in Thailand's Chi Watershed: Implications for Flood Management